# Usage and Temporal Patterns of Public Bicycle Systems: Comparison among Points of Interest

**Xingchen Yan** [1], **Liangpeng Gao** [2], **Jun Chen** [3,4,*] and **Xiaofei Ye** [5]

1 College of Automobile and Traffic Engineering, Nanjing Forestry University, Nanjing 210037, China; xingchenyan.acad@gmail.com
2 Institute of Transportation, Fujian University of Technology, Fuzhou 350118, China; liangpenggao.acad@gmail.com
3 School of Transportation, Southeast University, Nanjing 211189, China
4 National Demonstration Center for Experimental Road and Traffic Engineering Education, Southeast University, Nanjing 211189, China
5 School of Maritime and Transportation, Ningbo University, Ningbo 315211, China; yexiaofei@nbu.edu.cn
* Correspondence: chenjun@seu.edu.cn; Tel.: +86-139-1394-5222

**Abstract:** The public bicycle system is an important component of "mobility as a service" and has become increasingly popular in recent years. To provide a better understanding of the station activity and driving mechanisms of public bicycle systems, the study mainly compares the usage and temporal characteristics of public bicycles in the vicinity of the most common commuting-related points of interest and land use. It applies the peak hour factor, distribution fitting, and *K*-means clustering analysis on station-based data and performs the public bicycles usage and operation comparison among different points of interest and land use. The following results are acquired: (1) the demand type for universities and hospitals in peaks is return-oriented when that of middle schools is hire-oriented; (2) bike hire and return at metro stations and hospitals are frequent, while only the rental at malls is; (3) compared to middle schools and subway stations with the shortest bike usage duration, malls have the longest duration, valued at 18.08 min; and (4) medical and transportation land, with the most obvious morning return peak and the most concentrated usage in a whole day, respectively, both present a lag relation between bike rental and return. In rental-return similarity, the commercial and office lands present the highest level.

**Keywords:** public bicycle system; bicycle usage; temporal changes; *K*-means clustering; distribution model; pattern comparison

## 1. Introduction

The public bicycle system (PBS) is an important component of "mobility as a service" framework, and plays a key role in urban environment improvement, business upgrading, travel efficiency increase, traffic congestion relief, and transit service extension. As a green, low-carbon, active transportation mode suitable for short- and medium-distance travel, the public bicycle (PB) contributes to improve the urban air and travel environment [1]. It promotes the upgrading of the business model and improves travel efficiency by increasing neighborhood accessibility [2,3]. Convenient door-to-door transporting by PBS is helpful to reduce the use of individual cars and alleviate traffic congestion [4]. Because of its flexibility in connecting with various public transportation modes and expanding the coverage of public transport services, PBS could help to resolve the "last mile" problem [5].

The management of public bicycle resources (facilities and bicycles) is a routine for the transport authorities and PB operators. They benefit from a better understanding of PB station activity and driving mechanisms, such as riding demand and patterns generated from points of interest (POI) and urban land use in daily tasks [6]. Previous studies have shown that at the station level, public bicycle riding is associated with surrounding built

environment characteristics, such as population, job density, proximity to transit (subway and bus stations), bike lanes, and POIs (malls, parks, and restaurants) [7–13]. Maurer found that in Sacramento, California, neither population density nor bike lanes are significantly related to bicycle use, while in Minneapolis, there is a negative relationship between employment density and bicycle rental, with buses and rail being significant competitors to PBS [14]. According to a case study in Zhongshan, China [13], the number of other bicycle stations within a given catchment area (300 m) negatively affects demand. During the morning and evening peak hours, the number of land use types within the station buffer was associated with the highest positive impact on the demand and demand supply ratio. Specifically, there was no statistically significant effect of the public transportation variable, implying that the key role of public transportation is as a single mode for completing an entire trip, rather than an intuitive feeder mode. For PBS in Montreal, arrival and departure rates were positively correlated with both metro stations and population density within a 250 m buffer. In the arrival rate model, the opposite signs for morning and evening work densities highlight the potential use of the PBS for commuting. Furthermore, adding very high-capacity stations is not as usable as adding smaller stations [8]. El-Assi et al. found in Toronto that the number of bicycle stations in the vicinity was more highly correlated with the trip amount than dock number, and trip activity was higher in areas with universities as well as transit stations [15]. Other research has shown that an 11.5% increase in bike use was associated with a 1 km decrease in distance to the downtown centroid, while distance to the nearest station had a positive effect, possibly due to proximity reducing the use of individual stations. In addition, in an area with 1000 more jobs connected via transit, stations tended to have 0.8% more bicycle trips, although no significant effects were found, due to businesses [16].

Numerous studies on public bicycle systems are dedicated to exploring spatio-temporal characteristics, using station rental data as well as travel data. Kaltenbrunner et al. used an auto-regressive moving average forecasting technique, incorporating information from surrounding stations and prediction time interval to estimate bicycle usage [17]. However, they did not consider system attributes or the urban built environment. Based on usage patterns, Lathia et al. applied a hierarchical clustering algorithm to group stations in London to investigate the impact of access policy changes [18]. The study observed differences across policy changes, but examination of places around the changed stations (e.g., work establishments and residential areas) did little to explain why traffic changes occurred. Wu et al. explored the usage patterns of the PBSs, and to infer critical impact factors leading to different situations, they applied time series analysis on station-based data, and then compared the two systems by using a multinomial logistic regression model to better understand the relationship between public bicycle usage daily changing patterns and underlying spatial and cultural characteristics [19]. Zhao et al. estimated public bicycle daily trip characteristics, i.e., trip generation, trip attraction, trip distribution, and duration, using POI and smart card data from Nanjing, China. They examined the effect of the built environment on public bicycle usage with developed negative binomial regression models [20]. The research team of Yanjie Ji published two comparison studies between docked and dockless bike sharing systems in 2020. The first compared their usage regularity and the determinants [21]. The results showed that "trips during morning and afternoon peak hours" were positively associated with the regularity of both docked and dockless bike-sharing usage, while the "Riding distance" variable presented a negative association. For the impact of the built environment, they found working, residential, and transit POIs promoted the usage regularity of both bike sharing systems. In the latter, they reported that the density of entertainment POIs showed a positive and negative effect on dockless and docked systems [22].

Many previous studies have used various models to examine the impact of bicycle infrastructure, land use, and built environment attributes on arrival and departure flows or trip demand. However, few studies have examined the characteristics and differences in public bicycle daily usage and change patterns under the levels of these factor at site level,

especially POIs and land use. It is the basics for PBS management and operations. A variety of studies, mostly in the west, such as American and European cities, have examined the characteristics of PBS, but many PBSs are widely deployed in Chinese cities and need to be studied. Currently, most research studies have not discussed different usage patterns based on big data analytics. The paper addresses these shortcomings and contributes to extract the useful facts for PBS daily management and operations.

The rest of this paper consists of five sections: Section 2 indicates the datasets and methods for PB station analysis; Section 3 performs the PB usage and operation comparison among different POIs; Section 4 presents the results of the PB station clustering and volume feature comparison; Section 5 proposes some implications for PB rebalancing and land use optimization. Finally, Section 6 provides the summary, main results, contributions and limitations of this paper.

## 2. Data and Methods

### 2.1. Study Area and Dataset

The study area of the paper is the central town of Nanjing, including six districts. The location and administration zoning of the six districts are shown in Figure 1. The area of the central town is 787.45 km$^2$ and 3.33 million population. As the capital of Jiangsu, Nanjing had 901 PB stations up to 2017. With the promotion of TOD, PBS has been widely used in recent years, playing an increasingly important role in city traveling. The PBS data were collected from Nanjing Public Bicycle Co., Ltd. The dataset includes two parts: the station information and trip data. The first includes the station ID, name, address, longitude and latitude. The second contains more than 834,551 trips from 16 January (Monday) to 20 January (Friday). The trip data cover citizen ID number, user ID, station numbers of trip origin and destination, and the rental and return time.

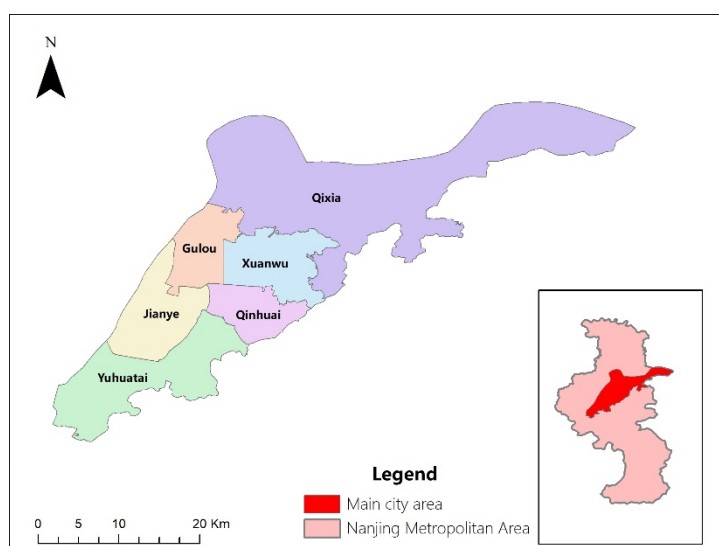

**Figure 1.** Location of study area in Nanjing and administrative zoning of main city area.

We also collected POI data and land use data. POI data presented the geographical location of urban facilities including malls, universities, schools, hospitals etc., obtained from open street map (https://www.openhistoricalmap.org/, accessed on 2 November 2021). The land use data were abstracted from the published Nanjing metropolitan area land status map (2017) by the Nanjing municipal bureau of planning and natural resources. The main types of land use for the study were commercial, office, residential, educational, transportation and medical land. According to the trip generation manual [23], these POIs and land use are the primary origins and destinations for commuting trips and thus, have the highest levels of travel demand in the central district of a city.

*2.2. Technology Pathway and Data Selection*

In this work, we aim to explore the usage and temporal patterns of public bicycles on the basis of PBS station data. By comparing operating characteristics of PB stations at or next to the selected POIs and types of land use, the PB usage and operational difference among POIs and land use are summarized. The technology pathway is shown in Figure 2. First, all the stations are filtered based on POIs, and the land use data and those stations related to the origins and destinations of commuting trip are selected. Second, the data of user arrival (including rental and return), usage duration and hourly volume calculated from the original data are analyzed, using peak hour factor and distribution fitting. Afterward, *K*-means clustering and the *L* method are applied to discover the station clusters, and the characteristics of the grouped stations are discussed.

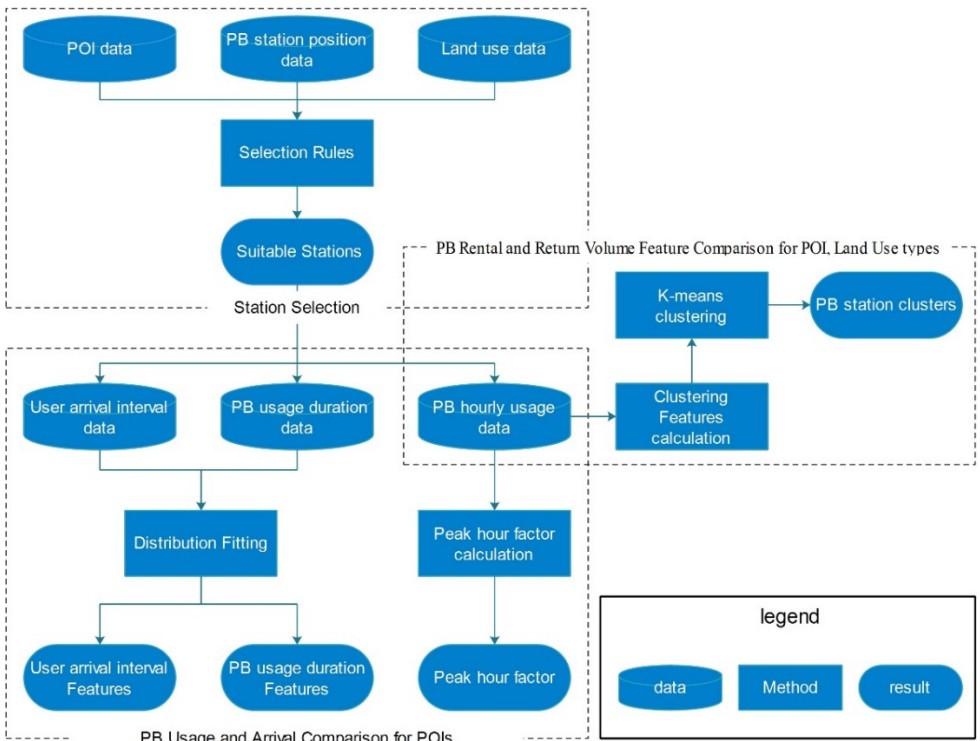

**Figure 2.** Technology pathway of the study.

The kernel density of stations is shown in Figure 3. To explore the influence of POIs on public bicycle usage, stations adjacent to schools, shopping malls, hospitals and rail transit stations were selected. Stations in or next to commercial, office, residential, educational, transportation and medical land were chosen to study the impact of land use on public bicycle usage. Based on the above criteria, the final number of stations selected was 300, most of which were distributed in the red and orange area in Figure 3. Obviously, the stations selected by the above two rules are partially the same.

Since the daily hiring of public bicycles is periodic, i.e., the vehicle borrowing and returning are basically the same every weekday in normal weather, the study finally selected the hiring data on Wednesday, 19 January 2017, the most stable day, through an initial comparison. Because the bike hiring is almost zero between 0:00 and 5:00, we chose 6:00 to 23:00 as the data analysis period.

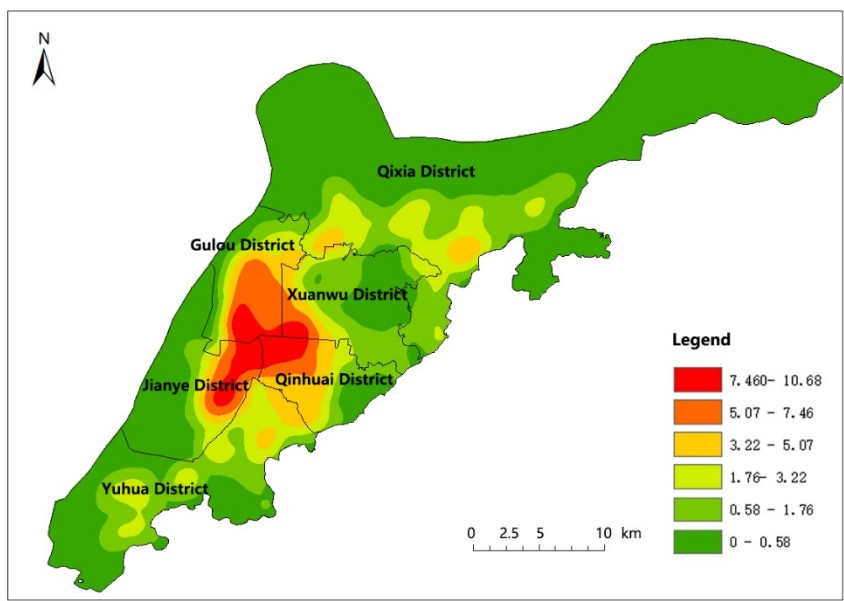

**Figure 3.** Kernel analysis for number of PBS stations.

### 2.3. Feature Selection for Clustering

The inventory level of a PB station is determined by the real-time bike hire and return. Various demands for different bike usage can lead to lack of empty docks for returning bikes or a lack of inventory to meet requests for renting bikes. Therefore, the time-varying patterns of PB usage, especially in peaks, are of importance. The bike inventory is greatly influenced by the quantitative and trend relationships between bike rental and return in these processes. To study the above time variability, similarity and trend correlation of bike hire and return of different types of POIs or land use, we proposed some indicators to measure the time-varying feature (see Figure 4a), similarity feature (Figure 4b) and lead–lag relationship feature (Figure 4c), respectively, in addition to the common statistics. Together, they were the inputting features for clustering later.

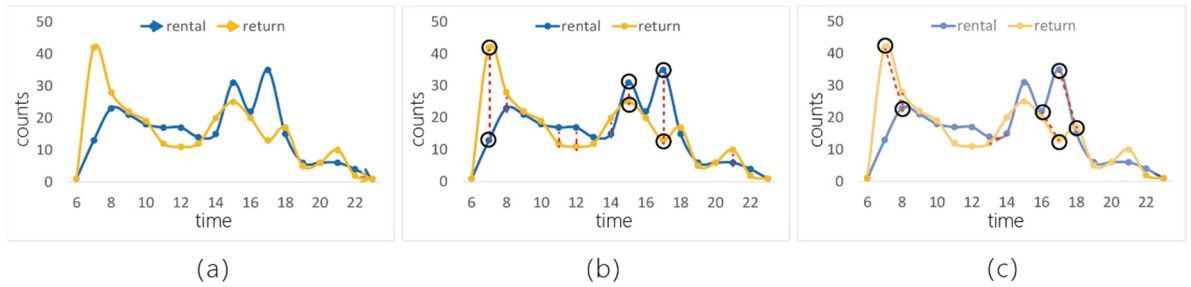

**Figure 4.** A sample of (**a**) time-varying feature, (**b**) similarity feature, and (**c**) lead–lag relationship feature for PB hourly checkout and check-in.

(1) Statistical Feature Variables for PB Hire and Return

The mean, median, standard deviation, maximum value, skewness, kurtosis, and bimodality coefficient were selected as the variables characterizing the hourly checkout and check-in of PB. The first four describe the average level, fluctuation and peak value, while the last three variables depict the shape of the distribution. Skewness is used to measure the asymmetry of the distribution of PB hiring data and is calculated as follows.

$$S = \frac{1}{n}\sum_{i=1}^{n}\left[\left(\frac{U_i - \mu}{\sigma}\right)^3\right] \tag{1}$$

where $n$ is the amount of hours for the analysis period; $U_i$ is the amount of hourly borrowed/returned vehicles (bikes/hour); and $\mu$ and $\sigma$ represent the mean and standard deviation of check-out/-in volume, respectively. When skewness < 0, the distribution is left skewed, while it is right skewed if skewness > 0. The data are relatively evenly distributed on both sides of the mean when skewness = 0.

Kurtosis is a measure of the steepness or "tailedness" of the probability distribution of a random variable. The kurtosis of a distribution is defined as follows:

$$K = \frac{1}{n}\sum_{i=1}^{n}\left[\left(\frac{U_i - \mu}{\sigma}\right)^4\right] \tag{2}$$

The symbols have the same meaning as above. Usually, the kurtosis value is subtracted by 3, also known as the excess kurtosis, so the kurtosis value of the normal distribution is equal to 0. When the kurtosis value > 0, it means that the data distribution is steeper, compared with the normal distribution, and when the kurtosis value < 0, it means that the data distribution is flatter, compared with the normal distribution.

The bimodality coefficient [24], which measures the multimodality of a statistical distribution, i.e., whether a distribution follows a single distribution or a multivariate distribution, is calculated as follows:

$$BC = \frac{S^2 + 1}{K + 3 \cdot \frac{(n-1)^2}{(n-2)(n-3)}} \tag{3}$$

The symbolic meaning is the same as before, and the critical multimodal coefficient $BC_{\text{crit}} = 0.555$, which is less than or equal to this value for a single distribution and greater than for a multivariate distribution.

(2) Time-varying feature variables for PB hire and return

To characterize the shape of the public bicycle borrowing and returning time-varying curve, the public bicycle hiring moments are considered data points, and the skewness, kurtosis and bimodality coefficient of the bike hiring time-varying curve are defined as follows:

$$S_t = \frac{1}{n_t}\sum_{i=1}^{n_t}\left[\left(\frac{T_i - \mu_t}{\sigma_t}\right)^3\right] \tag{4}$$

where $S_t$ is the skewness of the borrowing/returning curve. When the curve skewness < 0, borrowing and returning is biased toward the evening peak; when skewness = 0, it means that the distribution is even in the morning and evening; and when skewness > 0, borrowing and returning is biased toward the morning peak. $n_t$ is the total number of borrowing/returning moments; $T_i$ is the borrowing/returning moments ($T_i$ = 6–23); and $\mu_t$ and $\sigma_t$ are the mean and standard deviation of the borrowing/returning moments, respectively.

$$K_t = \frac{1}{n_t}\sum_{i=1}^{n_t}\left[\left(\frac{T_i - \mu_t}{\sigma_t}\right)^4\right] \tag{5}$$

where $K_t$ is the kurtosis of the PB borrowing/returning curve; other symbols have the same meaning as before.

$$BC_t = \frac{S_t^2 + 1}{K_t + 3 \cdot \frac{(n_t-1)^2}{(n_t-2)(n_t-3)}} \tag{6}$$

where $BC_t$ is the bimodality coefficient of the PB borrowing/returning curve, where less than 0.555 means that there is only one peak period and greater than 0.555 means that there are multiple peak periods.

(3) Similarity Feature of PB Hire and Return

To investigate the similarity of hourly public bicycle hire and return at stations, the cosine similarity, dynamic time warping cosine similarity (*DTWCS*) and dynamic time

warping distance (*DTWD*) are adopted and defined. The equation of cosine similarity is as follows:

$$CS = \frac{\vec{X} \cdot \vec{Y}}{|X| \cdot |Y|},$$

(7)

where $\vec{X}$, $\vec{Y}$ respectively, represent the time series of the PB hire/return hourly volume.

Dynamic time warping (DTW) is a well-known technique to find an optimal alignment between two given (time-dependent) sequences under certain restrictions [25]. The study adopted DTW to align the PB hire and return time series of a station. The *DTWCS* and *DTWD* are computed as below:

$$DTWCS = \frac{\vec{X}' \cdot \vec{Y}'}{|X'| \cdot |Y'|}$$

(8)

$$DTWD = \sum_{i=1}^{n} |x_i - y_i|$$

(9)

where $\vec{X}'$ and $\vec{Y}'$ represent the time series of the PB hire/return hourly volume after dynamic time warping, respectively, and $x_i$ and $y_i$ represent the vector components of the time series of borrowed/returned PBs after DTW, respectively.

(4) Lead–lag Relationship Feature Variables between PB Hire and Return

The lead–lag relationship is the phenomenon where a certain time-series lags behind and partially replicates the movement of the leading time-series. It is generally applied in financial markets [26]. Here, we use the lead–lag relationship to investigate the correlation of PB hire and return volume time series. The DTW gives the indices of two aligned time series, which are monotonically increasing sequences. The aligned indices have three relationships: less than, equal to and greater than. The components of the first time series, having smaller indices, means that it leads the latter, while the other two represent synchronization and lagging. Therefore, the difference of the corresponding indices is used to illustrate the three relationships. Based on that, the total lead–lag coefficient, synchronization rate, leading rate and lagging rate are defined to characterize the overall relationship between the PB hire and return volume sequences. The calculation formulae are as follows:

$$LL = \sum_{j=1}^{n} (ix_j - iy_j),$$

(10)

$$r_s = \frac{n_s}{n},$$

(11)

$$r_{lead} = \frac{n_{lead}}{n},$$

(12)

$$r_{lag} = \frac{n_{lag}}{n},$$

(13)

where $ix_j$ and $iy_j$ represent the coordinate indexes of the time series of PB hire and return volume series after DTW; and $n_s$, $n_{lead}$, $n_{lag}$, and $n$ represent the number of synchronous, leading, lagging and total amount of the aligned indices, respectively.

### 2.4. K-Means Clustering

For PB stations distributed at different types of POIs and land use, it is arbitrary and time consuming to average their features individually and then make a pairwise comparison. For this situation, cluster analysis provides a more efficient way, and it is the organization of a collection of patterns into classes based on similarity. Among the techniques, *K*-means is one of the dominantly used data mining algorithms [27–29]. It is very popular for data clustering, which aims at the local minimum of the distortion [30,31].

*K*-means has better clustering fitness than the others when considering performance in time complexity and the influence of the data type, size, and number of clusters.

(1) K-means clustering

*K*-means clustering is the most widely used partitional clustering algorithm [32]. The goal of *K*-means clustering is to partition n points (which can be one observation or one instance of a sample) into *K* clusters such that each point is assigned to one cluster of which the centroid is the closest to it based on the particular proximity measure chosen. The following is an outline of the basic *K*-means algorithm:

Step 1: Select *K* points as initial centroids.
Step 2: Form *K* clusters by assigning each point to its closest centroid.
Step 3: Recompute the centroid of each cluster.
Step 4: Repeat Steps 2–3 until the convergence criterion is met.

In the third step, a wide range of proximity measures can be used while computing the closest centroid. The choice can significantly affect the centroid assignment and the quality of the final solution. The different kinds of measures which can be used here are the city block distance, Euclidean distance, correlation distance, and cosine similarity.

(2) Determining the optimal number of clusters

The study uses the *L* method based on the evaluation graph [33] and silhouette coefficient to determine and validate the optimal number of clusters, respectively.

The information required to determine an appropriate number of clusters/segments to return is contained in an evaluation graph that is created by the clustering/segmentation algorithm. The evaluation graph is a two-dimensional plot, where the *x*-axis is the number of clusters, and the *y*-axis is a measure of the quality or error of a clustering consisting of *x* clusters. The *y*-axis values in the evaluation graph can be any evaluation metric, such as distance, similarity, error, or quality.

Figure 5 shows an example of an evaluation graph. From Figure 5a, we can see that the data points of the evaluation graph have three significantly different regions: a steep zone on the left, a flat zone on the right, and a gradient zone in the middle. The optimal number of clusters is obtained in the asymptotic region as in Figure 5b, so it is only necessary to find a point in the asymptotic region and use that point as a divider to perform a linear fit to the data points in the steep region on the left and the data points in the flat region; The number of clusters is obtained when the best fit to the data points in both regions is obtained at the same time. Now define a metric that captures the interpolated mean of the mixed root mean squared error of the two region fits, as in Equation (14):

$$RMSE_e = \frac{c-1}{b-1}RMSE_c(L_c) + \frac{b-c}{b-1}RMSE_e(R_c) \tag{14}$$

where *RMSE(Lc)* is the root mean squared error of the best-fit line for the sequence of points in *Lc* (and similarly for *Rc*). The weights are proportional to the lengths of *Lc* (*c*−1) and *Rc* (*b*−*c*). *b* is the maximum pre-set number of clusters, i.e., the maxima of the *x*-axis. We seek the value of *c*, such that RMSE is minimized, that is, the following:

$$c^\wedge = \underset{c}{\operatorname{argmin}} RMSE_c \tag{15}$$

Besides the *L* method, the silhouette coefficient is another good indicator of the quality of clustering, which combines the cohesion and separation of clusters [34]. The value is in the range of −1 to 1; the larger the value, the better the clustering effect.

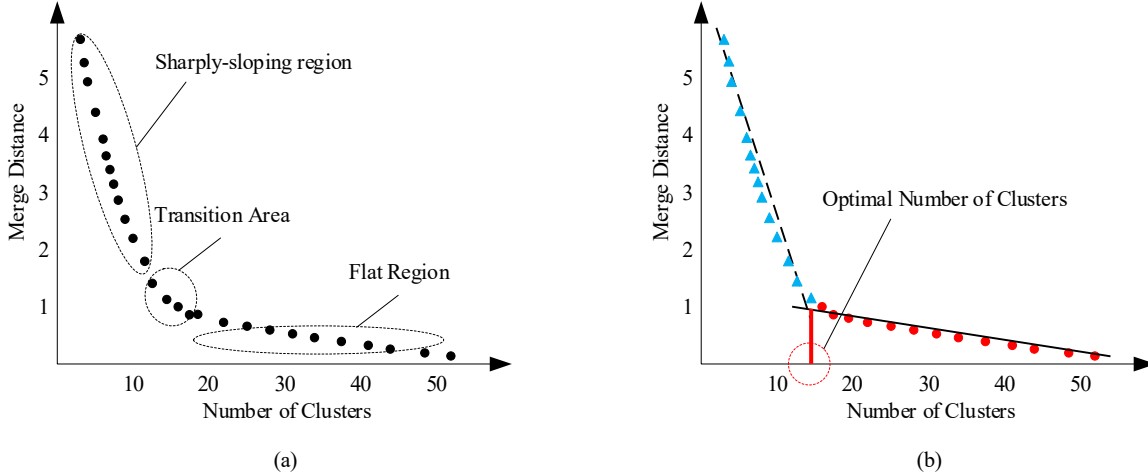

**Figure 5.** A sample evaluation graph.

## 3. PB Usage and Operation Comparison among Different POIs

### 3.1. Peak Hour Factor for PB Usage

Like the peak hour factor in the traffic volume study, we define the PB peak hour factor for bike hire and return to explore the characteristics of PB usage in rush hours. The equations are shown below.

$$PHF_h = \frac{V_h}{V_{ah}}, \tag{16}$$

$$PHF_r = \frac{V_r}{V_{ar}}, \tag{17}$$

where $V_h$ and $V_r$ are the PB hire and return volumes during peak hour, and $V_{ah}$ and $V_{ar}$ represent the accumulated hire and return volumes all day.

Based on the peak hour factor for each type of POI (see Table 1), the stations at hospitals have the biggest peak hour factor for bike return, at 0.151 followed by universities of 0.134. Thus, the stations at universities and hospitals are return oriented. On the contrary, travelers prefer using PB in their departure from middle schools. During peak hours, the PB hire and return at malls and metro stations are basically balanced with a difference of less than 0.01. This point was confirmed by the subsequent kurtosis for the PB rental and return volume time series in transportation and commercial land.

**Table 1.** Peak hour factors for different types of POI.

| POI Type | Hire | Return | Feature Description |
|---|---|---|---|
| University | 0.104 | 0.134 | return oriented |
| Middle school | 0.116 | 0.104 | hire oriented |
| Mall | 0.102 | 0.111 | hire–return balanced |
| Hospital | 0.117 | 0.151 | return oriented |
| Metro station | 0.115 | 0.113 | hire–return balanced |

### 3.2. Distribution for User Arrival Interval

Based on the arrival data of PB hire and return in the vicinity of five types of POI, the negative exponential distributions were obtained using least squares, and the results of the distribution fitting for each type of stations are shown in Table 2.

**Table 2.** Fitting results of arrival interval distribution of PB stations at different POIs.

| Type of POI | Hire Arrival Interval | | | | | | | Return Arrival Interval | | | | | | |
|---|---|---|---|---|---|---|---|---|---|---|---|---|---|---|
| | $\lambda$ (Counts/min) | Mean (min) | 85th PV [1] (min) | SSE [2] | $R^2$ | Adj $R^2$ | RMSE [3] | $\lambda$ (Counts/min) | Mean (min) | 85th PV (min) | SSE | $R^2$ | Adj $R^2$ | RMSE |
| University | 0.245 | 4.078 | 7.737 | 0.004 | 0.920 | 0.920 | 0.016 | 0.290 | 3.446 | 6.537 | 0.003 | 0.949 | 0.949 | 0.015 |
| Middle school | 0.291 | 3.438 | 6.522 | 0.009 | 0.847 | 0.847 | 0.026 | 0.245 | 4.077 | 7.734 | 0.004 | 0.920 | 0.920 | 0.016 |
| Mall | 0.588 | 1.702 | 3.228 | 0.063 | 0.856 | 0.856 | 0.047 | 0.320 | 3.127 | 5.932 | 0.008 | 0.886 | 0.886 | 0.023 |
| Hospital | 0.420 | 2.382 | 4.518 | 0.015 | 0.930 | 0.930 | 0.023 | 0.624 | 1.603 | 3.042 | 0.072 | 0.847 | 0.847 | 0.050 |
| Metro station | 0.654 | 1.530 | 2.902 | 0.014 | 0.952 | 0.952 | 0.022 | 0.458 | 2.182 | 4.139 | 0.013 | 0.942 | 0.942 | 0.021 |

[1] 85th percentile value. [2] The sum of squares due to error. [3] Root mean squared error.

We can see that metro stations have the densest PB hire arrival, with an arrival rate at 0.654 counts/min and 1.530 min of average interval, the shortest among the five types of POI. Malls and hospitals follow closely behind, while middle schools and universities are the bottom two, with more than 3 min of interval mean. In respect to return arrival, hospitals and metro stations are the top two types of POI, with 1.603 min and 2.182 min, respectively, both below 3 min. The other three are all over the duration. Overall, the PB hire and return at metro stations and hospitals are frequent, while only the rental at malls is dense.

### 3.3. Distribution Features for PB Usage Duration

From Table 3, the PB usage around malls has the longest duration at 18.08 min, while the durations in the buffer areas of middle schools and metro stations are the shortest. The 85th percentile values of duration indicates that most use PB for less than 30 min while the maximum durations are no more than 100 min.

**Table 3.** Statistics of vehicle usage duration at PB stations at different POIs.

| Type of POI | Mean (min) | Median (min) | 85th PV [1] (min) | Std [2] | Min [3] (min) | Max [4] (min) |
|---|---|---|---|---|---|---|
| University | 17.00 | 13.07 | 27.17 | 14.93 | 1.67 | 88.22 |
| Middle school | 15.85 | 10.37 | 25.72 | 15.09 | 1.37 | 82.77 |
| Mall | 18.08 | 12.51 | 31.46 | 16.91 | 1.02 | 96.23 |
| Hospital | 16.71 | 11.40 | 27.87 | 15.65 | 1.00 | 99.43 |
| Metro station | 15.90 | 10.76 | 27.80 | 15.04 | 1.12 | 98.13 |

[1] 85th percentile value [2] Standard error. [3] Minimum. [4] Maximum.

To fit the duration data, we used 15 continuous distributions, including Birnbaum–Saunders, exponential, gamma, generalized extreme value (GEV), generalized pareto, inverse Gaussian, logistic, loglogistic, lognormal, Nakagami, normal, Rayleigh, Rician, t-location scale, and uniform. The fitting results for the duration distributions of the five types of POI are shown in Figure 6a–e and Table 4. For all types of POI, the interval with the highest percentage and the best-fit distribution are consistent. It shows that the 4–6 min interval covers the biggest share in comparison to the others. GEV and loglogistic are always the top two best-fit distributions. Specially, the fitted GEV with the parameter k over zero are Type II GEV of which the tails decrease as a polynomial.

**Table 4.** Probability distribution parameters and goodness-of-fit test results.

| Type | Name | Parameters | LL [1] | KS [2] | AIC [3] | AICc [4] | BIC [5] |
|---|---|---|---|---|---|---|---|
| University | gev | $k$: 0.36, $\sigma$: 7.71, $\theta$: 8.96 | −1440 | Y | 2885 | 2885 | 2897 |
| | loglogistic | $\mu$: 2.47, $\sigma$: 0.57 | −1446 | Y | 2896 | 2896 | 2904 |
| Middle School | gev | $k$: 0.44, $\sigma$: 6.81, $\theta$: 6.96 | −1239 | Y | 2254 | 2254 | 2265 |
| | loglogistic | $\mu$: 2.24, $\sigma$: 0.68 | −1126 | Y | 2257 | 2257 | 2264 |
| Mall | gev | $k$: 0.42, $\sigma$: 7.90, $\theta$: 9.03 | −7238 | Y | 14,482 | 14,482 | 14,499 |
| | loglogistic | $\mu$: 2.49, $\sigma$: 0.58 | −7251 | Y | 14,506 | 14,506 | 14,517 |
| Hospital | gev | $k$: 0.43, $\sigma$: 7.16, $\theta$: 8.65 | −4215 | Y | 8436 | 8436 | 8451 |
| | loglogistic | $\mu$: 2.44, $\sigma$: 0.54 | −4220 | Y | 8444 | 8444 | 8455 |
| Metro station | gev | $k$: 0.45, $\sigma$: 6.87, $\theta$: 7.95 | −4861 | Y | 9728 | 9728 | 9744 |
| | loglogistic | $\mu$: 2.36, $\sigma$: 0.57 | −4868 | Y | 9741 | 9741 | 9751 |

[1] Log-likelihood of the model on the dataset. [2] Kolmogorov–Smirnov test statistic. [3] Akaike information criterion. [4] Bias-corrected information criterion. [5] Bayesian information criterion.

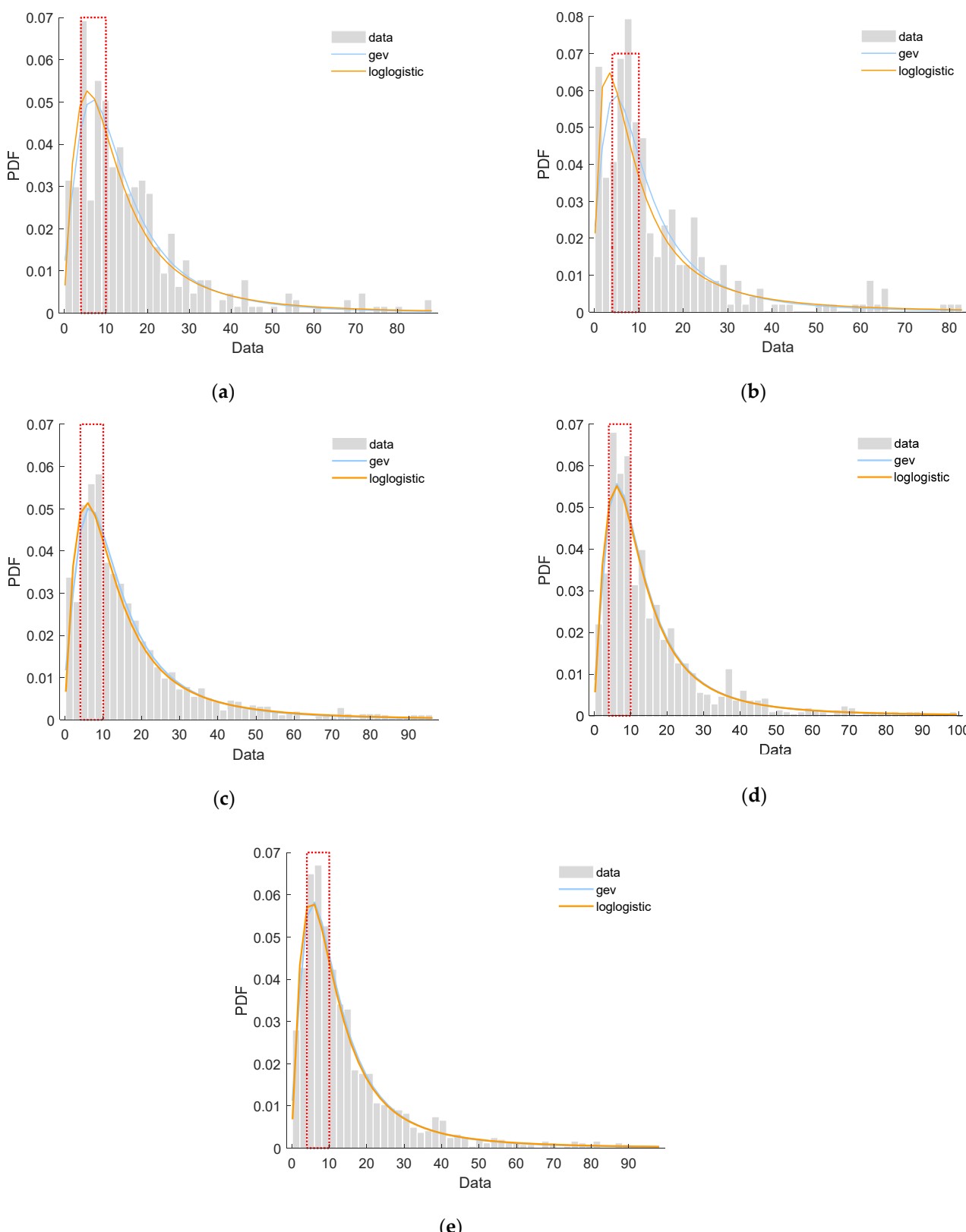

**Figure 6.** (**a**) Fitting results for universities, (**b**) middle schools, (**c**) malls, (**d**) hospitals, (**e**) metro stations.

## 4. PB Rental and Return Volume Feature Comparison among Different Types of POIs and Land Use

### 4.1. Clustering Results for PB Stations

For PB stations distributed in or next to the selected types of POIs and land use, the four types of feature variables selected in Section 2.3 were taken as the input of *K*-means clustering, and the optimal number of clustering was determined by the *L* method as illustrated in Figure 7. The silhouette coefficient was to validate the clustering results as shown in Figure 8.

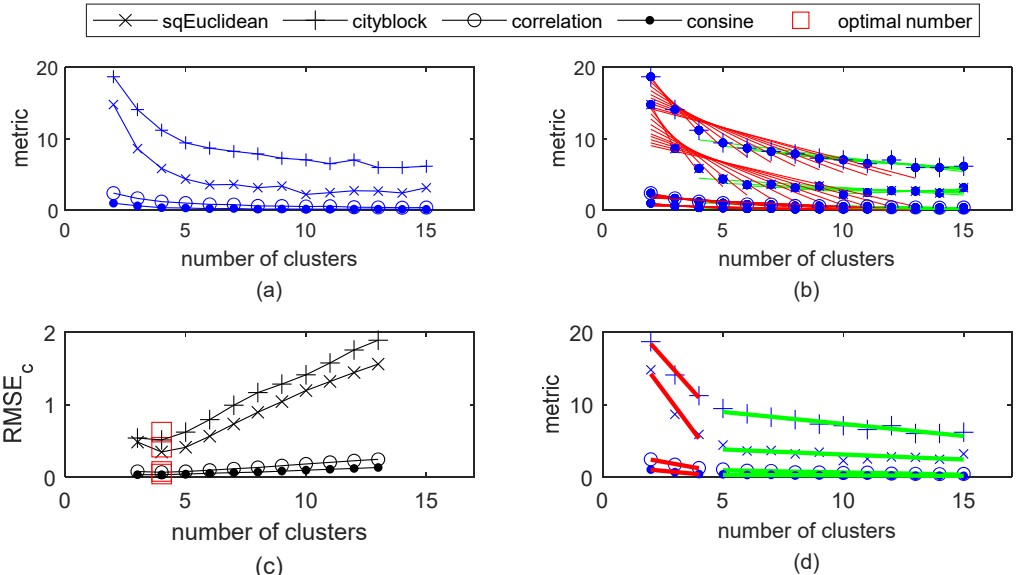

**Figure 7.** (**a**) Evaluation graph, (**b**) possible fitting lines, (**c**) RMSE, and (**d**) best-fit lines for city-block distance, Euclidean distance, correlation distance, and cosine similarity.

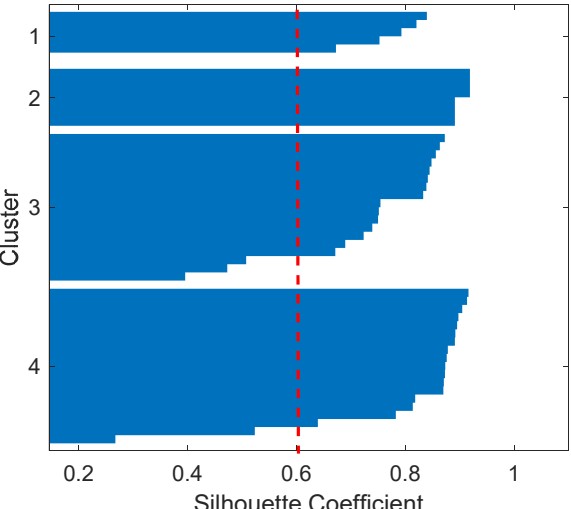

**Figure 8.** Silhouette coefficient values of clustering = 4.

Figures 7 and 8 show that the optimal number of clusters for 300 sites is 4 classes, and most of the silhouette coefficients of the 4 classes are greater than 0.6, proving that the clustering effect is good.

### 4.2. Rental and Return Feature Comparison among Different Types of Land Use

The main types of POIs and land use characteristics of the four clusters are summarized in Figure 5. Based on Table 5, the statistical feature, time-varying feature, similarity feature and lead–lag relationship feature are analyzed for the four types of rental stations below.

**Table 5.** Full-day hire and return characteristics of four clusters of PB rental stations.

| Cluster | | 1 | | 2 | | 3 | | 4 | |
|---|---|---|---|---|---|---|---|---|---|
| Main Type of POIs and Land Use | | Medical | | Transportation | | Commercial, Office | | Residential, Education | |
| Account (Counts) | | 34 | | 51 | | 88 | | 127 | |
| Usage Type | | Rental | Return | Rental | Return | Rental | Return | Rental | Return |
| Statistical Feature | Means (counts/h) | 15.33 | 14.83 | 14.36 | 13.89 | 24.92 | 25.92 | 9.03 | 9.58 |
| | Median (counts/h) | 15 | 13.5 | 12.5 | 11.5 | 27 | 27 | 9 | 9 |
| | Std [1] | 8.92 | 10.38 | 10.85 | 10.89 | 15.26 | 17.13 | 5.61 | 5.78 |
| | Max [2] (counts/h) | 35 | 41 | 43 | 35 | 55 | 58 | 20 | 22 |
| | S [3] | 0.49 | 1.02 | 1.11 | 0.79 | 0.38 | 0.47 | 0.5 | 0.48 |
| | K [4] | −0.74 | 0.66 | 0.67 | −0.72 | −1.16 | −1.19 | −0.96 | −0.72 |
| | Bc [5] | 0.51 | 0.53 | 0.57 | 0.66 | 0.56 | 0.61 | 0.56 | 0.5 |
| Time-varying Feature | $S_t$ [6] | 0.05 | 0.24 | −0.02 | −0.02 | 0.11 | 0.09 | 0.1 | 0.04 |
| | $K_t$ [7] | −0.97 | −1.1 | −0.73 | −0.73 | −0.95 | −1.02 | −1.05 | −1.06 |
| | $Bc_t$ [8] | 0.49 | 0.55 | 0.45 | 0.44 | 0.5 | 0.51 | 0.51 | 0.51 |
| Similarity Feature | Cs [9] | 0.86 | | 0.92 | | 0.95 | | 0.88 | |
| | Dtwcs [10] | 0.9 | | 0.93 | | 0.95 | | 0.92 | |
| | Dtwd [11] | 7.66 | | 5.4 | | 3.79 | | 7.01 | |
| Lead–lag relationship Feature | LL [12] | 10.5 | | 3.5 | | −4 | | −1.5 | |
| | $R_s$ [13] | 0.55 | | 0.82 | | 0.78 | | 0.62 | |
| | $R_{lead}$ [14] | 0.07 | | 0 | | 0.17 | | 0.23 | |
| | $R_{lag}$ [15] | 0.39 | | 0.18 | | 0.05 | | 0.15 | |

[1] Standard error. [2] Maximum. [3] Skewness. [4] Kurtosis. [5] Bimodality coefficient. [6] Skewness of borrowing/returning volume time series curve. [7] Kurtosis of borrowing/returning volume time series curve. [8] Bimodality coefficient of borrowing/returning volume time series curve. [9] Cosine similarity. [10] Dynamic time warping cosine similarity. [11] Dynamic time warping distance. [12] Total lead–lag coefficient. [13] Synchronization rate. [14] Leading rate. [15] Lagging rate.

(1) Statistical feature for PB hourly hire and return

The means and median of PB volume data indicate that the four clusters of stations are basically balanced in terms of bike rental and return for a whole day, among which commercial and office sites have the largest volume and volatility, residential and educational sites have the smallest, and the other two categories are in the middle. Moreover, the kurtosis of the hire/return volume also confirms the fluctuation difference of the four clusters from the other side. The hiring and returning volume of all four types of rental sites is right skewed, which means that the hourly amounts of PB usage are mainly small- and medium-sized, and the peak hours of PB rental are short. The bimodality coefficients display that both PB rental and return for the stations in transportation, commercial and office land present a bimodality feature, with uniform characteristics in medical. With respect to the residential and educational land, the PB rental and return are different: bimodal and unimodal, respectively.

(2) Time-varying feature for PB hourly hire and return

Overall, half of the skewness values being close to zero (<0.1) indicates that the morning and evening peaks are nearly the same in most cases. The return flow at PB rental sites in medical land has the most obvious morning peak (skewness = 0.24), followed by the rentals in commercial, office (0.11) and residential, education (0.1). Notably, only the PB

volume time series in transportation land present as slightly left skewed (skewness < 0). It means that PB usage in the morning is less than the second half of a day. We can see all the bimodality coefficients are below 0.555, which means the volume time series of four clusters are unimodal. The biggest kurtosis of cluster 2 proves that the PB usage in transportation land is the most concentrated among the four clusters.

(3) Similarity feature of PB hourly hire and return

The cosine similarity, DTWCS and DTWD together show that the PB rental and return time series of the commercial and office land have the highest similarity, while the similarities for transportation, residential, education and medical land decrease in order.

(4) Lead–lag relationship feature between PB hourly hire and return

In total, the PB rentals for medical and transportation lag behind the returns, with 10.5 and 3.5 total lead–lag coefficients, respectively, while commercial, office, residential and education land present a leading relationship. It means that the travelers are in favor of PB in their arrivals to hospitals and transportation stations, while the groups in the other types of land prefer PB in their departures. In more than half of the time, PB rentals and returns are synchronized for all the types of land use (see $R_s$ row). Through comparing the leading rate and lag rate data, we find that the lead–lag relationship for residential and education land presents a mixed feature. The lead–lag coefficient for this type of land use is smaller than that of commercial, office land. However, it has a higher leading rate. This can be explained by its lagging rate. With a lagging rate of 0.15, no longer a low level, the PB rental at the stations lags behind the return in some parts of a day. It narrows the leading difference in total.

## 5. Implications for PB Rebalancing and Mixed Land Use

The PBS imbalance is caused when 'tidal flows' of bike sharing trips move from or to certain areas of a city, such as from residential to commercial zones during the morning peak hour [35]. Meanwhile, residential suburbanization and jobs–housing separation trend in Chinese cities worsens the issue [36]. To address this problem, fleet balancing or reallocation and land use optimizing are the most used measures from the dynamic and static aspects, respectively [36,37]. The results of the paper provide some workable directions for the two measures.

From the calculation of the peak hour factor for PB usage, we found that the usage patterns of middle schools and hospitals were different, being hire-oriented and return-oriented, respectively. The PBS operators may perform the dedicated bike dispatch actions between middle schools and hospitals to balance the two POIs quickly. The volume comparison among different types of POIs and land use shows that the PB stations in medical land have the most obvious morning return peak, while those in commercial, residential and education present a significant morning hire demand. However, the result of the peak hour factor indicates that the PB usage near malls is balanced when commercial land has the highest level in hire–return similarity and synchronization rate. Therefore, it is possible to mix medical and residential land to achieve a PB resource that is balanced within a community or reduce fleet allocation significantly.

## 6. Conclusions

In the paper, we mainly compared the usage and temporal characteristics of PB in the vicinity of the most common commuting-related POIs and land use and acquired the differences among them. First, those stations adjacent to the common POIs and land use were selected. Afterward, the user arrival, usage duration and hourly volume calculated from the original data were analyzed, using peak hour factor and distribution fitting. Finally, *K*-means clustering and the *L* method were applied to discover the station clusters, and the characteristics of the grouped stations were discussed. The following results and conclusions were obtained:

- The PB demand types for universities and hospitals in peak hours are return oriented while that of middle schools is hire oriented. For malls and metro stations, it is hire–return balanced.
- The PB hire and return at metro stations and hospitals, with an average arrival interval less than 3 min, is frequent while only the rental at malls is.
- In PB usage, malls have the longest duration at 18.08 min, while those of middle schools and metro stations are the shortest. For all types of POI, 4–6 min interval covers the biggest share, and Type II GEV and loglogistic are the most suitable distributions for usage duration.
- Commercial and office land have the largest PB volume, while residential and educational have the smallest. Medical and transportation land, with the most obvious morning return peak and the most concentrated usage in a whole day, respectively, both present a lag relation between bike rental and return. In rental–return similarity, the commercial and office land present the highest level.

The usage and operating characteristics of public bicycle concluded by the paper provide valuable knowledge for urban authorities and public bicycle operators in deploying public bicycle resources. The fitted distributions for user arrival interval and usage duration will be helpful in the public bicycle studies of other researchers, e.g., operating simulations and theoretical derivations. Due to privacy restrictions on data, the socioeconomic attributes of public users are not included in this paper. Further in-depth research is necessary in the future when the relevant data are licensed.

**Author Contributions:** L.G. undertook the data collection and analysis. X.Y. (Xingchen Yan) provided an interpretation of the results and wrote the majority of the paper. X.Y. (Xiaofei Ye) contributed to the paper review and editing. J.C. was the supervisor of the paper. All authors have read and agreed to the published version of the manuscript.

**Funding:** This research was funded by the Natural Science Foundation of Jiangsu Province (grant No. BK20180775), Key Project of National Natural Science Foundation of China (grant No. 51638004), Natural Science Foundation of Zhejiang Province (grant No. LY20E080011&LY21E080008), the Fujian Natural Science Foundation (grant No. 2020J05194) and the Technology Program of Fujian University of Technology (grant No. GY-Z19094,GY-Z17155), and Basic public welfare research project of Zhejiang Province 2018 (grant No. LGF18E090005).

**Institutional Review Board Statement:** Not applicable.

**Informed Consent Statement:** Not applicable.

**Data Availability Statement:** Restrictions apply to the availability of these data. Data was obtained from Nanjing Public Bicycle Co., Ltd. and are available from the authors with the permission of Nanjing Public Bicycle Co., Ltd.

**Acknowledgments:** The authors would like to express their sincere thanks to the anonymous reviewers for their constructive comments on an earlier version of this manuscript.

**Conflicts of Interest:** The authors declare no conflict of interest.

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
