# Peer review of "Usage and Temporal Patterns of Public Bicycle Systems: Comparison among Points of Interest"

_information, doi:10.3390/info12110470_

Round 1

Reviewer 1 Report

The content of this paper is very detailed and the results are novel. It would be best if the authors add discussion on the research results, which can improve the practical significance of the paper.

Author Response

Dear Professor,

We are truly grateful to your critical comments and thoughtful suggestions. Based on these comments and suggestions, we have made careful modifications on the original manuscript. All changes made to the text are highlighted in yellow. We hope the new manuscript will meet your requirements. Below you will find our point-by-point responses to your comments:

  1. A discussion was added as Part 5, in which the implications of the results were described.

Thank you again for your time and consideration.

Sincerely,

Authors

Reviewer 2 Report

Review paper “Usage and Temporal Patterns of Public Bicycle Systems: Comparison among Points of Interest and Land Use”

General comment

  1. The use of the English language must be improved.
  2. The title and the text of the manuscript should be revised to remove “land use” since the analysis are only focused on points of interest. To include “land use” issues and other types of analysis must be addressed.
  3. The paper is well structured, but it would the interesting to have some information on how this data could be used in real-life problems, i.e., showing examples about its possible application in future works

Introduction

  1. Nowadays, the number of dockless and electric bike-sharing services have grown as MaaS, it would be interesting to include these topics in the introduction and compare them with the dock systems that are presented in this section
  2. Also, would it be interesting to contrast and compare the differences between points of interest according to the type o bicycle rental (regular bicycles vs. e-bikes; dock-less bicycles vs. docked services)
  3. It would be important to identify similarities between different bicycle-sharing services and the outcomes for POIs
  4. The last two paragraphs must be increased to better understand the objectives of the paper, as well as the novelty of this research.

Data and Methods

  1. What is the “technology pipeline” in line 123?
  2. In figure 3, the legend could be improved to be better seen and understandable, clarifying the meaning of the “number ranges”
  3. In figure 4, all axes need to be labelled
  4. Why the time-varying characteristics (similarity feature, and lead-lag relationship) were chosen? And an explanation must be made to understand this choice.
  5. Furthermore, a justification about the suitability of the used method for this study must be presented.
  6. k-means clustering explanation must be better explained to make clearer the link to the subject of this paper, instead of the provided generic explanation.

PB usage and operation comparison among different POIs

  1. Why only universities, middle schools, malls, hospitals and metro stations were chosen as POIs?
  2. In line 291 the word “including” appears twice

PB rental and returns feature comparison among different types of land use

  1. This point must be merged with the previous point (comparison among different POIs) since the approach is mainly focused on POIs being the relation to land use extrapolated from the POIs instead of a specific and separated analysis
  2. After the results of the study, it would be interesting to know how it could affect urban planning and PB stations distribution. How this data could help real-life problems on PB assignments?

A Discussion section should be provided to better understand the novelty of this work and the improvements concerning the state of the art on this issue.

Author Response

Dear Professor,

We are truly grateful to your critical comments and thoughtful suggestions. Based on these comments and suggestions, we have made careful modifications on the original manuscript. All changes made to the text are highlighted in yellow. We hope the new manuscript will meet your requirements. Below you will find our point-by-point responses to your comments:

  1. According to your suggestions, “land use” was removed from the paper title.
  2. The latest comparison studies were reviewed and added in introduction part, please see line 85-92.
  3. The “technology pipeline” was updated with “technology pathway” (see line 132).
  4. Figure 3 and 4 were improved according to your kind suggestions.
  5. An explanation for choosing time-varying characteristics (similarity feature, and lead-lag relationship) was added. Please see line 154-162.
  6. The reason for selecting K-means was explained in line 231-238.
  7. We selected these POIs and types of land use according to Trip generation manual which was indexed as a reference in the new version.
  8. Your advice about merging Part 4 to Part 3 is spot on. But, it is too long for Part 3. Instead, we revised the subtitles of the two parts to reflect their contents accurately.
  9. A discussion was added as Part 5, in which the implications of the results were described.
  10. Due to the short time, we only corrected the partial grammar errors in the manuscript. We will use English editing service afterwards.

Thank you again for your correction. We know that every submission is a valuable learning opportunity, your guidance has greatly improved our paper.

Sincerely,

Authors